# The oldest Pottery Neolithic (PN) culture of northeastern Iran: First absolute dating from eastern Mazandaran plains

Rahmat Abbasnejad Seresti [1]*, Xinying Zhou[2‡], Seyyed Kamal Asadi Ojaei[3]

1 Department of Archaeology, Faculty of Arts and Architecture, University of Mazandaran, Babolsar, I.R. Iran, 2 Key Laboratory of Vertebrate Evolution and Human Origins of Chinese Academy of Sciences, Institute of Vertebrate Paleontology and Paleoanthropology, Beijing, China, 3 Department of Archaeology, Faculty of Arts and Architecture, University of Mazandaran, Babolsar, I.R. Iran

◔ These authors contributed equally to this work.
‡ XZ also contributed equally to this work.
* r.abbasnejad@umz.ac.ir

## Abstract

In the past, establishing a clear chronology for the Epipalaeolithic and Neolithic periods in eastern Mazandaran proved challenging. A major obstacle had been the lack of radiocarbon dating. Previous dates provided by Coon and McBurney were not considered reliable, even after recalibrations. However, over the last fifteen years, new archaeological fieldwork and research have significantly enhanced our understanding of these periods. Recent excavations at the PN sites of Touq Tappeh and Tappeh Valiki have provided new information about the Epipalaeolithic and Neolithic chronology and dating. The sites yielded the oldest dating of the PN in northeastern Iran so far, making the PN of eastern Mazandaran start at least from the first half of the 7th millennium BC and lasted until the early 6th millennium BC (c. 6600–5800 BC). While Tappeh Valiki represents the oldest dates, the PN periods may have started in the region even earlier, given the presence of potteries from the lowest layers of the site. Analysis of the available material from these sites through dating indicates strong regional connections, while also showing inter-regional connections. The new dating from the old and new Epipalaeolithic and Neolithic sites of eastern Mazandaran suggests there is no gap between them, which is not surprising given the favorable environment during the early Holocene.

## 1. Introduction

Decades after the cave excavations by Carlton Stevens Coon and Charles McBurney, studies of the Epipalaeolithic and Neolithic periods on the eastern plains of Mazandaran resumed at the beginning of the 21st century. These studies have been ongoing, with caves being re-excavated. The excavation at Touq Tappeh in 2020 and Tappeh Valiki in 2022 has brought greater attention to the significance, capabilities,

provided the original author and source are credited.

**Data availability statement:** All data are in the paper and/or Supporting information file.

**Funding:** The author(s) received no specific funding for this work.

**Competing interests:** No authors have competing interests The authors have declared that no competing interests exist.

and challenges of the PN on the plains. The excavations and ongoing research have raised important questions about the formation of the sites, their dating, technologies (including pottery and lithics), paleoclimate, Caspian Sea level changes, and the impact on the past communities. Understanding the region's Neolithization process, food production, and domestication should be a priority for further research. Studying the absolute chronology and dating of the prehistoric sites on the Neka and Behshahr plains form the basis for understanding the abovementioned problems and issues.

The plains of eastern Mazandaran have been hypothesized as a pathway for the spread of the Neolithic lifestyle from the eastern wings of the Fertile Crescent to Western Central Asia, which may have influenced the development of Neolithic cultures such as Djeitun [1,2]. To further test this hypothesis, it is essential to establish absolute chronologies for Neolithic sites in eastern Mazandaran. However, apart from Coon's works, absolute dating from the Neolithic contexts of the region has not been reported. Information regarding the origins and development of technologies such as pottery is limited to a few publications [3,4,5]. Since pottery can serve as a valuable indicator for comparative studies of archaeological layers at sites, having absolute dating for the PN layers in the region would be beneficial in determining the chronology of technological innovations and subsistence.

Previously, three Neolithic pottery horizons were considered for this region: the Caspian Neolithic Software in the middle of the 7th millennium BC; the Djeitun pottery that appears contemporary with the previous type in the late 7th to the middle of the 6th millennium BC; and the black-on-red pottery at the end of the 6th millennium BC, which has been considered as an indicator of the transition from Neolithic to Chalcolithic [6,4,7]. Before the excavations of Tappeh Valiki and Touq Tappeh, our understanding of the PN in the eastern Mazandaran plains was limited to surface surveys. These excavations allow us to understand better the region's chronology for the PN, along with technological developments, especially pottery and lithic.

## 2. Geography and environment

The geographical area of this research includes the southeastern basin of the Caspian Sea, particularly the Behshahr and Neka plains. These plains are the natural extension of the western part of the Gorgan Plain [8] and constitute almost an independent region along the northern slopes of the central Alborz mountains. Geologically the plain has a gentle slope with a maximum 25km width between the Alborz foothills and the sea shores, located between the mouth of Neka River to the north and the delta of the Gorgan Rive to the east, and the Sari Plain to the west. The Gorgan Plain has been considered as an area of dust during leoss formation, and its extent depends on the transgression and regression of the Caspian Sea level in the late Quaternary [9].

Harris [10] described this area as part of Northeastern Iran and West Central Asia. The region has abundant natural resources. Most of the mountains are covered with Hyrcanian forests, which provide a habitat for animals such as deer, gazelle, boar, etc. The plains have fertile soil suitable for crops such as rice, wheat, barley, and citrus orchards, and the sea provides many food sources such as fish, oysters, shells, and seals. The Caspian Sea has been strongly affected by climate changes, and the

environment and human life have been affected by its transgressions and regressions over the time. The most important developments of the Caspian Sea in Neolithic include the transgressions of the Neo-Caspian and the regressions of 8.4~8.2 kyr at early Holocene [11,12,13,14].

## 3. Neolithic field work background

Coon's excavations at the cave sites Hotu and Kamarband in Behshahr in the middle of the 20th century initiated the research on the Epipalaeolithic and Neolithic on the eastern plains of Mazandaran. He excavated 28 layers in the Kamarband Cave (divided into four cultural horizons) and 17 layers in the Hotu Cave. The cultural layers of these neighboring caves share great similarity, both including remains from the Epipalaeolithic, Pre-Pottery Neolithic (PPN), PN, Chalcolithic, disturbed Iron Age, historical and Islamic periods [6,15,16]. In the 1960s, Charles McBurney excavated and reported 23 cultural layers in Al Tappeh Cave, all belonging to the Epipalaeolithic [17,18]. After this initial research, there was a four-decade hiatus in prehistoric research of the region until the field surveys of the Behshahr and Neka plains [19] and sounding at Touq Tappeh in Neka [20]. Following this, excavation at the Komishan Cave revealed in-situ Epipalaeolithic layers, while the Neolithic and later layers were greatly disturbed by mining activities [21,22]. Through another surface survey project in the region, 16 Neolithic sites have been reported [23,24]. Excavation at the terrace in front of Komishan Cave, known as the Komishani open site, provides new dating for the Epipalaeolithic and PPN [25,26]. The recent stratified excavations of Touq Tappeh [27] and Tappeh Valiki [28], as the first field research in the eastern Mazandaran plains, are the most recent fieldwork in the region. Additionally, the re-excavation of Hotu and Kamarband caves has produced new findings on dating, stratification, animal and plant remains, etc [29,30]. During the latest field survey, the number of PN sites in the region increased to over 40, additionally, for the first time, several Neolithic sites were recorded and introduced in the highlands of eastern Mazandaran [31,3].

## 4. Material and methods

The excavations of Touq Tappeh took place in 2020 and Tappeh Valiki in 2022; the proposal for excavations first requested by the University of Mazandaran, with contribution of the Cultural Heritage, Handicrafts, and Tourism Organization of Mazandaran; the permission for field work were provided by Iranian Research Institute of Cultural Heritage and Tourism, as well as the Iranian Center for Archaeological Research. Touq Tappeh (N36º 42' 41.90", E53º 20' 54.78") and Tappeh Valiki (N36º 42' 57.74", E53º 17' 29.64") are both located on the Neka plain, at a distance of 5 km from each other and 16 km to the south of the Caspian Sea shores, and 8 km to the north of the Central Alborz foothills (Figs 1 and 2). Both sites are situated -6m above sea level and 4–5 meters above the surrounding lands. Touq Tappeh was excavated in the summer of 2020, while Tappeh Valiki was excavated in the spring of 2021. The aims and objectives of both excavations were to investigate the Neolithization process, the beginning of food production, the evolution of technologies, and the impact of climate change on prehistoric communities in the southeast region of the Caspian Sea. At Touq Tappeh we opened two trenches (TT1 with dimensions of 1×4 meters, and TT2 with dimensions of 2×3 meters) to understand the site`s stratigraphy. A total depth of 220 cm out of the 4-meter stratigraphy at this site belong to the PN. At Tappeh Valiki, four trenches were opened, i.e., Tr1, Tr2, Tr3, and Tr4 with dimensions of 2×4, 1×4, 3×2, and 2×5 meters. At this site, except for a thin layer of the historical and Chalcolithic periods, the rest of the layers of about 200 cm in depth, all belong to the PN.

Seven and five samples for radiocarbon dating were selected from trenches of Touq Tappeh, and Tappeh Valiki respectively. All the samples were sent to the Xi'an Center for Accelerator Mass Spectrometry of Institute of Earth Environment, Chinese Academy of Sciences. First, they were cleaned of any adhering sediment and other material and then crushed. This was followed by an acid–base–acid (ABA) chemical pretreatment, which consisted of washing in hot HCl (5%), rinsing, treatment with 1% NaOH, rinsing again, and then treatment with hot HCl (5%) and rinsing and drying. The pretreated samples underwent combustion to $CO_2$ by oxidation at 800 °C using CuO as the catalyst. And then purification of $CO_2$

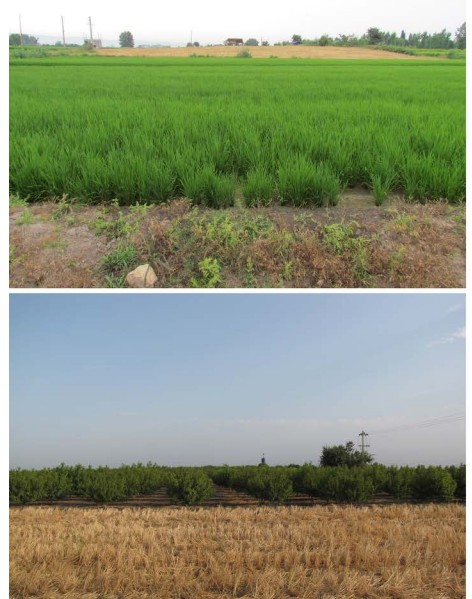
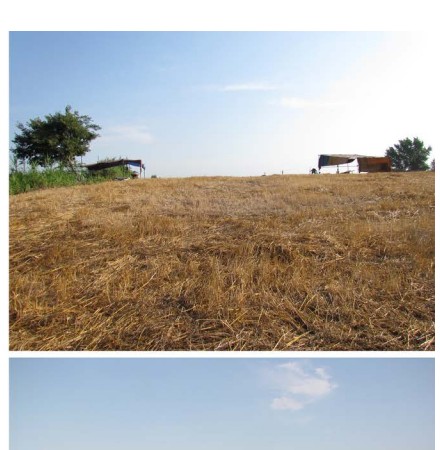
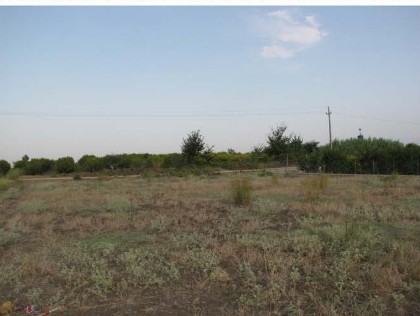

**Fig 1. General view of Touq Tappeh (up) and Tappeh Valiki (down).**

in the presence of silver wire to absorb any SOx and NOx produced. and reduction of purified $CO_2$ to graphite with $H_2$ at 550 °C using an iron catalyst. The pressed graphite targets were then sent to the Radiocarbon Dating Laboratory. The radiocarbon ages and related information, including age calibration, were performed using the OxCal version 4.4 program for accurate interpretation of the results (Fig 3).

## 5. Results: The old and new chronologies, and related issues

Hotu and Kamarband caves were some of the earliest ancient sites in Iran to be dated using the C14 method. The existing absolute dating of the Epipalaeolithic and Neolithic in the eastern Mazandaran can be categorized into three groups:

1) The dating of charcoal samples obtained from the excavations of Hotu and Kamarband caves [32] is the first absolute dating in the region. However, these dating faces problems such as non-scientific and selective methods of recovering and recording materials, as well as "the wide range of statistical variability associated with early 'conventional' radiocarbon dating techniques" [7], which was unreliable despite recalculation and calibration (Table 1).

2) The calculation and recalibration of the radiocarbon dates of these caves, which was carried out based on the review of Coon's unpublished notes and plans (Table 1), has shown that the Epipalaeolithic at Hotu started from about 14000–11300 BC and at Kamarband around 13200–11000 BC (The new dating from the Kamarband Cave have not been officially published yet). The end of Epipalaeolithic is about 8000 BC and 7600 BC respectively at these two sites. Also, according to this dating, the PPN starts in 7940 and ends in 6465 BC and the PN starts in 7140 and ends in 5050 BC; [7,32,33]. The occupation at Al Tappeh Cave, with all of its layers belong to the Epipalaeolithic [18], was dated to 10991–11510 BC after recalibration (calibrated by Calib Rev 8.1.0, based on dating provided by Coon for Kamarband Cave and McBurney for Al Tappeh Cave [34]).

3) The new dating of the sites in the region has opened a new chapter on the Epipalaeolithic and Neolithic research and removed the time gaps between the two periods. According to the new dating (2-sigma), the Epipalaeolithic at Hotu Cave

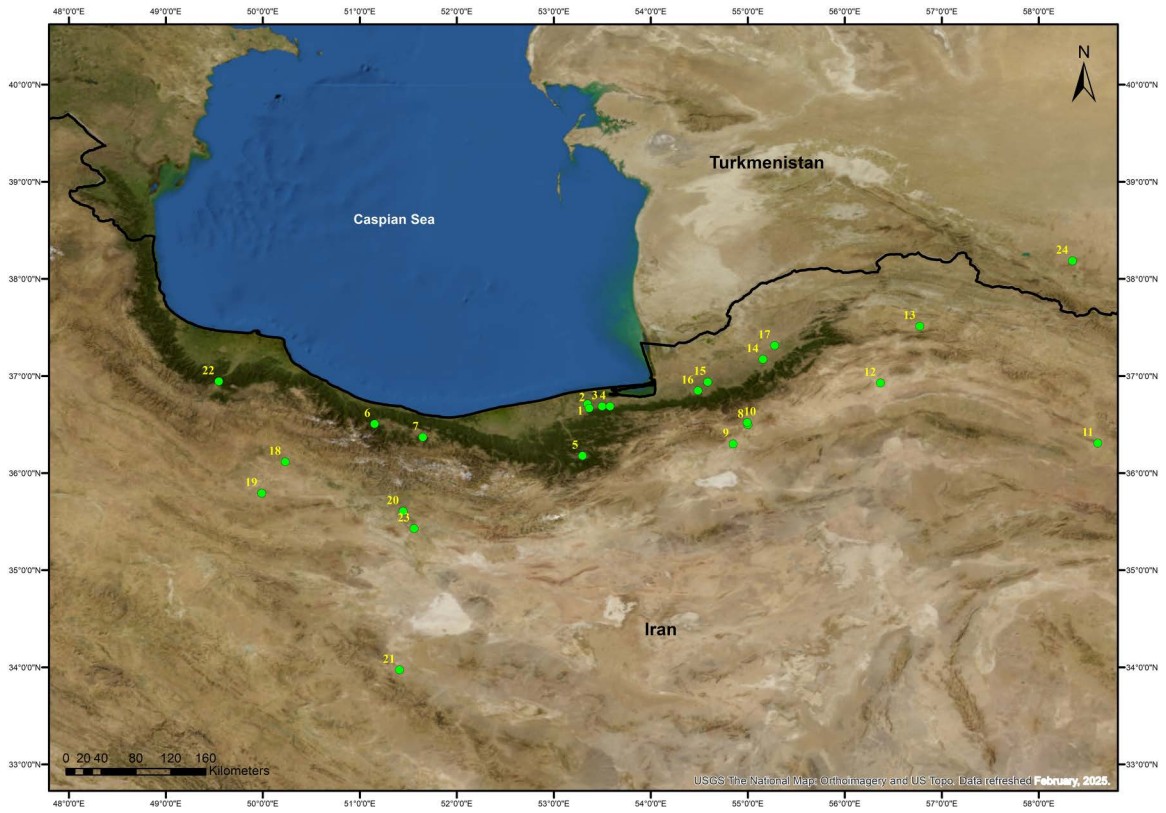

**Fig 2. Important Neolithic sites of Southeastern Caspian Sea and Adjacent Regions: 1) Komishan Cave and Komishani open site; 2) Touq Tappeh and Tappeh Valiki; 3) Hotu and Kamraband caves; 4) Al Tappeh Cave; 5) Qale`Pey; 6) Rashak III cave; 7) Eshkul Cave; 8) Sang-e Chakhmaq; 9) Kalateh Khan and Rouyan; 10) Deh Kheir; 11) Baluch; 12) Pahlavan; 13) Qale Khan; 14) Yarim; 15) Tureng; 16) Pookerdvall; 17) Aq Tappeh; 18) Ebrahim Abad; 19) Chahar Boneh 20) Cheshmeh Ali 21) Sialk 22) Shahran 23) Pardis 24) Djeitun.**

began at c. 11945–11800 BC and ended at c. 8130–7960 BC. The oldest PPN layers are dated to be c. 7653–7948 BC. The PN started from c. 6449–6351BC [35]. Two C14 dating samples from the Epipalaeolithic layers of Komishan Cave have presented dates around 12069–10632 BC. Since the Neolithic layers of this cave were extremely disturbed, the PPN has been identified only through the study of lithic assemblages [36,22]. The oldest date from the Komishani open site is c. 9256–9242 BC which belongs to the Epipalaeolithic. Also, the oldest PPN layer is dated 8634–8529 BC [14].

To sum up, the existing absolute dates were obtained only from the caves in the eastern Mazandaran, with no C14 dates of PN sites from the the plains. Respectively, seven charcoal samples from Touq Tappeh and five from Tappeh Valiki were selected for C14 dating. Based on these dates, the TT1 and TT2 in Touq Tappeh (Fig 4) respectively, show dates between 6250–6050 BC, and 6000–5800 BC; therefore, the PN at this site started in 6250 BC and ended in 5800 BC (Table 2). The PN layers of Tappeh Valiki have revealed older dates. Respectively, in trenches Tr3 and Tr4 (Fig 5) the dates obtained are between 6450–6100 BC and 6600–5900 BC; therefore, the PN at this site started in 6600 BC and ended in 5900 BC (Table 3). These dating show us that the phase of the PN of eastern Mazandaran, which contains Caspian Neolithic Soft-ware (The CNS), started at least from the mid-7th millennium BC and continued until the beginning of the 6th millennium BC.

Let's once again evaluate the PN dating of the Kamarband and Hotu caves by Ralph and the calculation and recalibration of the radiocarbon dates by Gregg and Thornton. As mentioned, the only PN dating in the region

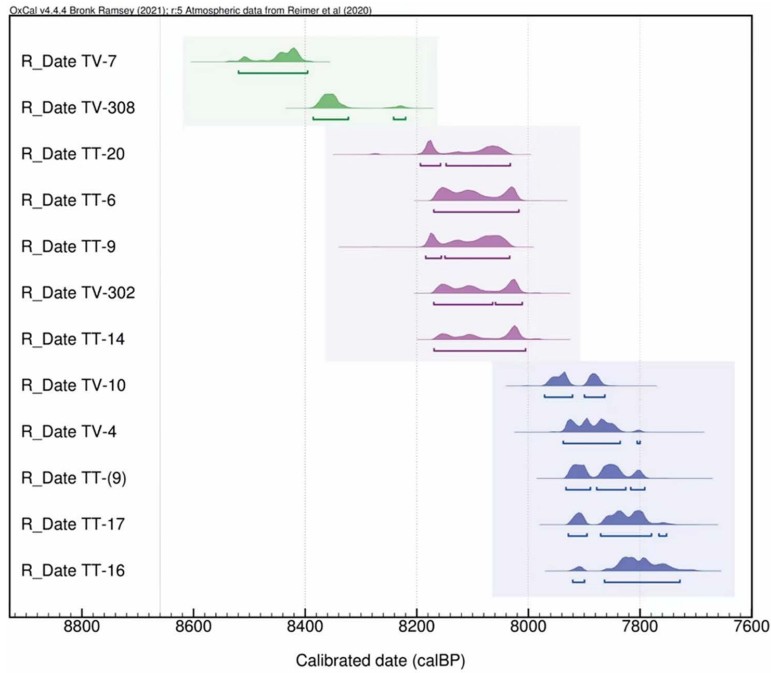

**Fig 3. Bayesian distributions of AMS C14 data, seven from Touq Tappeh (TT) and five from Tappeh Valiki (TV).**

**Table 1. Chronology of Epipalaeolithic and Neolithic of eastern Mazandaran based on Coon`s dating and calibration at 2012.**

| Period | The old dating (BP) [32] | | The new dating (BC | | Sample |
|---|---|---|---|---|---|
| | Hotu | Kamarband | Kamarband | Hotu | |
| The oldest level of Epi-Paleolithic | 12215±865 | 11900±775 | 13210-11000 | 13920-11350 | Charcoal |
| The oldest level of PPN | 8310±515 | 8140±490 | 7630-6465 | 7940-6650 | Charcoal |
| The oldest level of PN | 6575±440 | 7620±510 | 7140-6000 | 5975-5050 | Charcoal |

belongs to these two caves. The dates of the PN layers at Kamarband is estimated to be between 6610 and 5640 BC based on the calculation and recalibration [33], which are close to the dating from Tappeh Valiki (6566–5891 BC) and Touq Tappeh (6247–5774 BC). On the other hand, the new PN dating from Hotu Cave by de Groene and colleagues, based on a charcoal sample (6351–6449 BC), is worthy of consideration. However, given the difference in the thickness of occupation layers and the flourishing of PN sites, the authors believe that the PN on the plains of the eastern Mazandaran is older than the mentioned caves and probably started earlier on the plains. The more than 200 cm thickness of PN layers at Tappeh Valiki and Touq Tappeh, compared to 15 cm at Kamarband [7] and 25 cm at Hotu [29], along with the sudden increase in the number of PN sites to more than 40 sites [3], seem to support the abovementioned hypothesis. At present, no PPN site has been reported on the eastern Mazandaran plains, therefore, establishing the sequence of PPN and PN and confirming this hypothesis is somewhat difficult. So far, we do not have the exact dating of the end of the PN and the beginning of the Chalcolithic period, which is characterized by the black-on-red pottery. Based on the above argument and Tables 2 and 3, which contain new dates of the PN of East Mazandaran, the authors propose a chronological sequence for the Epipalaeolithic and Neolithic periods of the region as follows (Table 4).

**Fig 4. Stratigraphy section of Touq Tappeh trench TT2 (up) and TT1 (down).**

## 6. Discussion: Eastern Mazandaran at PN: Regional and inter-regional connections

### 6.1. Pottery

The dating of Touq Tappeh and Tappeh Valiki is considered the first step in tracing the origin of the PN in the eastern Mazandaran. In the past, the starting date of this period was established by dating the Hotu and Kamarband caves to around 6600 BC, however, due to errors in sampling and dating methods, this dating was considered unreliable. Upon re-excavation of Hotu and subsequent new dates, 6400 BC has been verified and accepted for the PN. As mentioned in Table 3, the PN on eastern Mazandaran plains is already almost 200 years older than the dates we have from the cave sites. It is worth noting that the settlement layers in these caves are reported to be 40 cm thick in total. This thickness, compared to the 220 cm thickness of the PN layers of Touq Tappeh and the 200 cm of Tappeh Valiki, strengthens the possibility that the plains were the origin of PN in the eastern Mazandaran and even in northeastern Iran.

**Table 2. C14 dating of Touq Tappeh (after calibrated with calib 8.1.0 [37]).**

| Lab No. | Sample No. | Material | Trench-Context (Depth) | Radiocarbon Age (BP) | 1-sigma Date BC | 2-sigma Date BC |
|---|---|---|---|---|---|---|
| XA57717 | TT-2020–6 | Charcoal | TT1- Con 13 (225 cm) | 7269±17 BP | 6204-6073 cal. BC | 6212-6069 cal. BC |
| XA57719 | TT-2020–9 | Charcoal | TT1- Con 16 (237 cm) | 7334±17 BP | 6233-6098 cal. BC | 6235-6094 cal. BC |
| XA57725 | TT-2020-14 | Charcoal | TT1- Con 19 (267 cm) | 7250±17 BP | 6197-6058 cal. BC | 6209-6050 cal. BC |
| XA57728 | TT-2020-20 | Charcoal | TT1- Con 22 (307 cm) | 7351±17 BP | 6242-6102 cal. BC | 6247-6093 cal. BC |
| XA57724 | TT-2020-(9) | Charcoal | TT2- Con 15 (233 cm) | 7022±17 BP | 5972-5884 cal. BC | 5984-5853 cal. BC |
| XA57726 | TT-2020-16 | Charcoal | TT2- Con 16 (292 cm) | 6973±17 BP | 5889-5824 cal. BC | 5908-5774 cal. BC |
| XA57727 | TT-2020-17 | Charcoal | TT2- Con 16 (315 cm) | 6997±17 BP | 5907-5842 cal. BC | 5972-5819 cal. BC |

The cultural interaction of eastern Mazandaran with adjacent regions is an important factor to consider regarding the origin of the PN in this region. The point of this interaction was the finding of two sherds that were obtained from the excavation of 2007 in Touq Tappeh, which were identified as Djeitun/Chakhmaq type [38]. In the WMSeC (Western Monud of Sang-e Chakhmaq), one sherd was found at level 1, and two at level 3 [39]. 10 dating samples from level 3 provided dates ranging from 8003±32–7900±30 B.P. [40]. After calibration using Calib Rev 8.1.0, the dates were determined to be 7045–6649 BC [37]. During the re-excavation of the WMSeC, two sherds were recovered in phase 3. Based on two dating samples, this phase was determined to date back to 7063–6827 BC [41]. Based on these sherds, the hypothesis of the PN origin of northeastern Iran from WMSeC (of which the eastern Mazandaran is also considered a part) has been proposed and dated to the beginning of the 7th millennium BC [41]. Eight sherds from Tappeh Rouyan, located 13 km southwest of Sang-e Chakhmaq [42] have been considered as a supporter and confirmation of the hypothesis. According to the mentioned hypotheses, the Chakhmaq culture has been divided into two phases and it has covered the Neolithic period of the entire northeastern Iran: The first phase is Aceramic/Proto-ceramic Chakhmaq, which is based on two sherds from phase three of WMSeC, and eight sherds from Tappeh Rouyan. The second phase is Ceramic Chakhmaq, based on the pottery sherds from EMSeC (Eastern Mound of Sang-e Chakhmaq). Based on the dates of 6800–6600 BC from trench 4 and 6000–5800 BC from trench 3 of Tepe Rouyan, the existing gap between the end of occupation at WMSeC and the beginning at EMSeC has been considered to be filled [42,43].

To evaluate the hypothesis mentioned above, it was formed only based on five pottery sherds recovered at the WMSeC; Masuda and Tsunki did not express a clear opinion about the beginning of the PN in WMSeC and only with vague and doubtful words, they believed the presence of these few sherds in the mentioned layers to be considered as an issue for future [39,40]. Roustaei's statement regarding the beginning of PN in phase 3 of WMSeC only based on two sherds isn't currently arguable [44,45]. If we consider the mentioned eight sherds from Tappeh Rouyan, which has only one dating to 6800–6600 BC, as a supporter for this hypothesis, it still faces challenges; Because a single dating sample (from Tappeh Rouyan) is not very reliable. In addition, the dating of the site is covered by Masuda's dating from the WMSeC, 7200–6600 BC, and does not cover any gap. Therefore, based on the available evidence, we cannot seriously rely on the two phases that have been proposed at the WMSeC. Because the dates of the EMSeC (6200–5400 BC), Kalateh Khan in Shahrud Plain (5600–5300 BC) and Deh Kheir (6000–5800 BC) in Bastam Plain [46,47] suggest the late 7th and 6th millennium BC for PN which is compatible with the dates of the Djeitun and its phases [1].

 

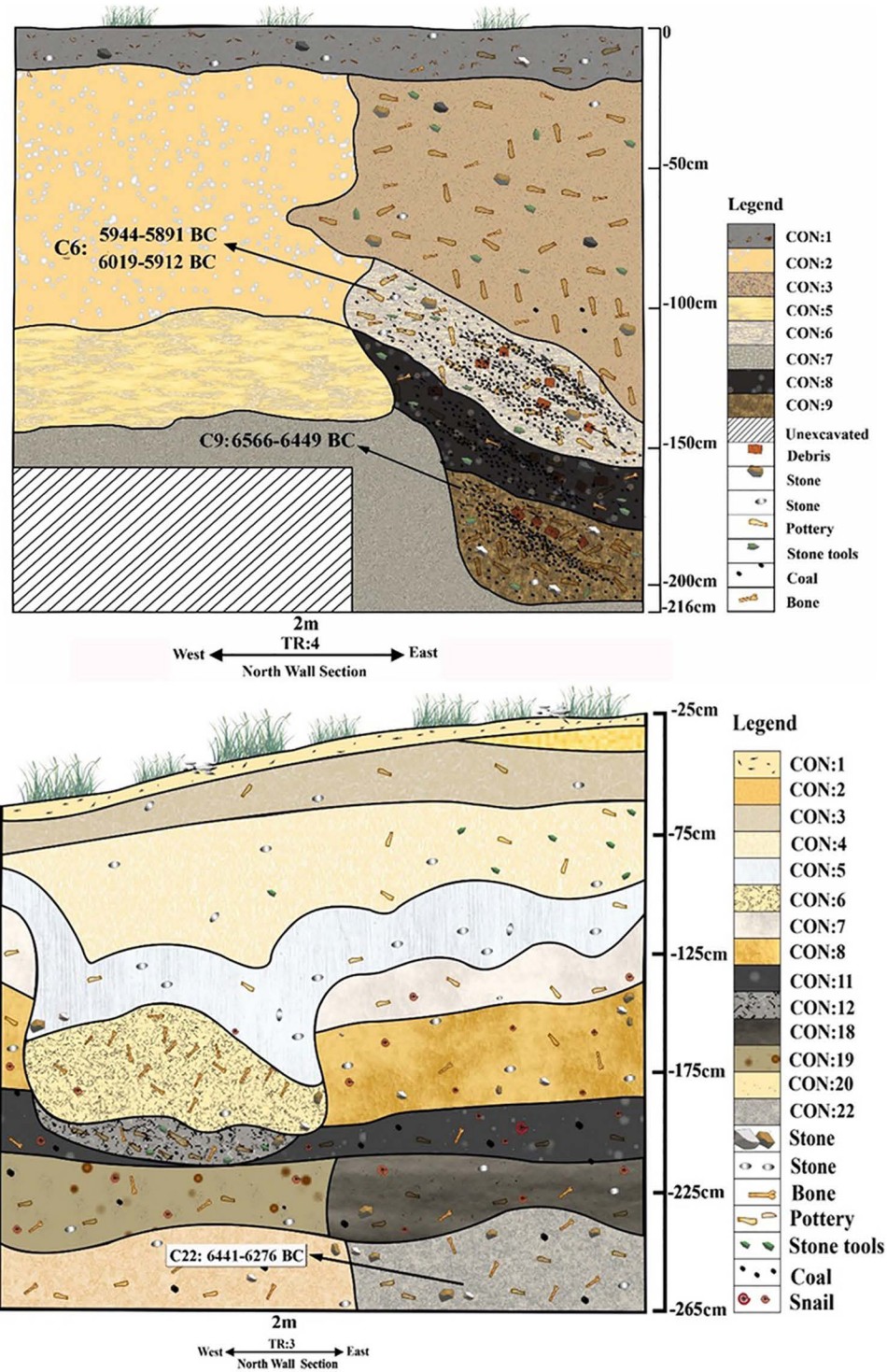

**Fig 5. Stratigraphy section of Tappeh Valiki trench Tr3 (up) and Tr4 (down).**

**Table 3. C14 dating of Tappeh Valiki (after calibrated with calib 8.1.0[37].**

| Lab No. | Sample No. | Material | Trench-Context (Depth) | Radiocarbon Age (BP) | 1-sigma Date BC | 2-sigma Date BC |
|---|---|---|---|---|---|---|
| XA57732 | TV-2022–410 | Charcoal | Tr4 – Con 6 (110 cm) | 7097 ± 19 BP | 6011-5925 cal. BC | 6019-5912 cal. BC |
| XA57730 | TV-2022–404 | Charcoal | Tr4 – Con 6 (130 cm) | 7048 ± 17 BP | 5975-5914 cal. BC | 5994-5891 cal. BC |
| XA57731 | TV-2022–407 | Charcoal | Tr4 – Con 9 (168 cm) | 7663 ± 18 BP | 6499-6457 cal. BC | 6566-6449 cal. BC |
| XA57733 | TV-2022–302 | Charcoal | Tr3 – Con 14 (178 cm) | 7258 ± 18 BP | 6201-6065 cal. BC | 6210-6060 cal. BC |
| XA57734 | TV-2022–305 | Charcoal | Tr3 – Con 22 (234 cm) | 7520 ± 18 BP | 6424-6392 cal. BC | 6441-6276 cal. BC |

**Table 4. Chronology table suggested by the Authors for Epipalaeolithic and Neolithic of Eastern Mazandaran.**

| Site Period | Hotu | Kamarband | Al Tappeh | Komishan | Komishani open site | Tappeh Valiki | Touq Tappeh |
|---|---|---|---|---|---|---|---|
| Epi-paleolithic | 12000-8000 BC | 14000-8500 BC | 11500-11000 BC | 12000-10600 BC | 9200-8600 BC | | |
| PPN | 8000-6700 BC | 7500-6000 BC | | Disturbed | 8600-8300 BC | | |
| PN | 6400-? BC | 6600-5600 BC | | | | 6600-5900 BC | 6250-5800 BC |

On the Gorgan Plain, absolute dating for the PN has not been reported; however, based on the pottery typology of Pookerdvall [48] and Aq Tappeh [49], it is suggested that they belong to the late 7th and 6th millennium BC. The oldest PN site in North Khorasan is Qale Khan, which dates back to 5800 BC [34,50]. Thus, the Djeitun pottery type is considered to be more recent than the CNS type. Moreover, Dyson included four sherds from the 1970s excavation of the WMSeC, as well as a few sherds he gathered during his visit to the site, undoubtedly as the CNS type [33]. Therefore, no PN cultures, older than the CNS culture, have been observed in northeastern Iran.

In 2020, two stratified trenches were excavated in Touq Tappeh. Trench TT1 with dimensions of 1 × 4 m at the site's highest point, was made and its PN layers thickness was 178 cm; 9.6 cubic meters of soil were excavated. TT2 with dimensions of 2 × 3 m was made in the center of the site and the PN layers thickness was 263 cm; the volume of soil excavated in this trench was 15.18 cubic meters. Of 2430 prehistoric pottery sherds, 1504 sherds belong to the Neolithic. Among the Neolithic pottery on this site, no sherd can be confidently attributed to the Djeitun/Chakhmaq type. Almost all of the sherds from Touq Tappeh belong to the CNS type described by Matson [5] and Dyson [4]. However, a few painted sherds found in Context 13 from the TT1 (6200–6100 BC) may indicate similarities with the Djeitun type. The reddish-orange grid motifs on the cream slip (Figs 6A2 and 7, sherd 1,10,12) were observed on the pottery of Pookerdvall on Gorgan Plain [48] and some other Neolithic sites of northeastern Iran. Also, pottery with black motifs on a pinkish-cream slip (Fig 6A2) that was reported in the pottery assemblages of the Kamarband [7] and Pookerdvall was also observed in Context 13. In other contexts of trench TT1, which have absolute C14 dating, almost all the sherds are simple (unpainted), which is one of the characteristics of the CNS type (Figs 6A1, B1-2, C1-2 and 8). The number of painted sherds obtained from the TT2 is more than that of the TT1 and their features are also more varied. The dating of Neolithic layers in TT2 is 6000–5800 BC. Two sherds from Context 16, TT2 are comparable to the *shady motifs* of the Djeitun type (Fig 7, sherds 6, 9). *Shady motifs* sherds have been recovered from EMSeC, Kalateh Khan, and Deh Kheir in the Shahrud and Bastam plains, and Aq Tappeh and Pookerdvall on the Gorgan Plain. The other sherds of this trench are the CNS type; The

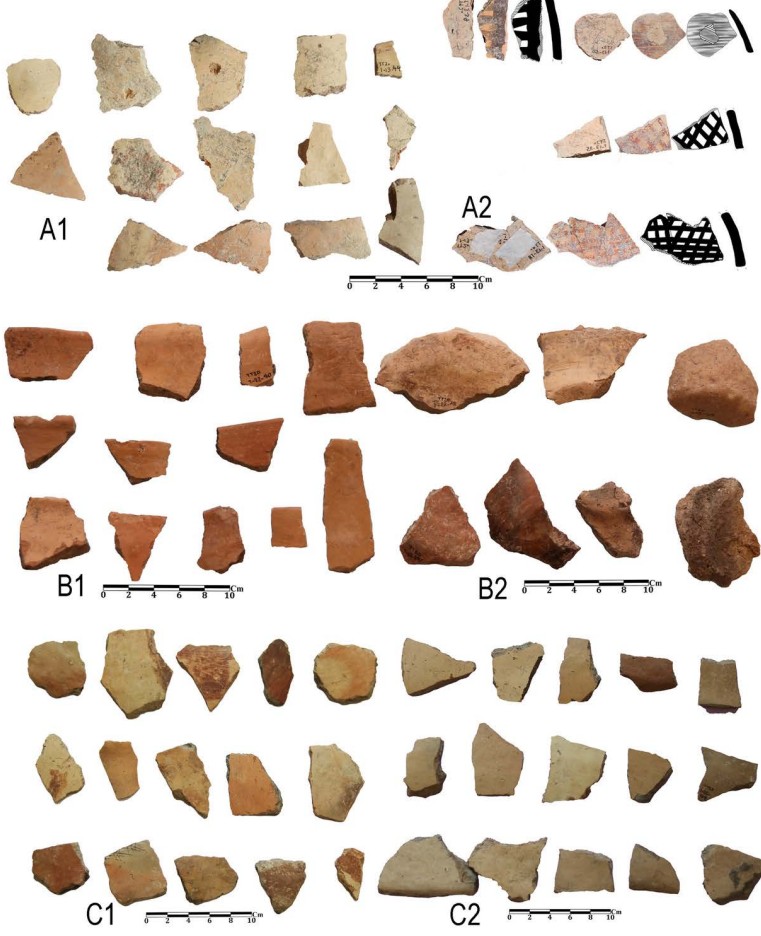

**Fig 6. Neolithic sherds of Touq Tappeh, trench TT1: Con 13 (A1, A2); Con 22 (B1, B2, C1, C2).**

*horizontal ladder motifs* on the rim of these sherds are one of the most significant and common motifs that have been found from this trench.

In the excavation of Tappeh Valiki, two main trenches were opened. The trench Tr3 is 2 × 3 m in the northern part of the site has PN layers of 1.4 m thick; 8.4 cubic meters of these layers were excavated. The trench Tr4 was created in the eastern part of the site with dimensions of 2 × 5 m and its PN layers' thickness was 2.16 m; the volume of excavated soil in this trench was 21.6 cubic meters. 1,641 sherds were obtained from three trenches. Of those, 1,247 sherds belong to the PN. Most of the sherds in this site indicate the CNS type features; however, three painted sherds are similar to the Djeitun/Chakhmaq types. Two sherds with *shady motifs* were recovered from context 6 of Tr4, which dated to 6000–5900 BC (Fig 10, sherd 4 and 5). Most of the Tr3 sherds' slips were ruined; however, one sherd with small colored circles was recovered from this trench. This sherd was found in Context 21, which is parallel to Context 22, dated to 6400–6300 BC (Fig 11, sherd 4). Sherds with Similar motifs have been found in the Djeitun culture sites, such as Djeitun, Togolok, Chupan, Pesijik, and Bami. According to Coolidge, this motif belongs to phase 2 of Djeitun culture, i.e., the mid-6th millennium BC (Middle Djeitun). The mentioned motif has been observed in Sialk I (5800–5300 BC), the EMSeC (6200–5700~5200 BC), and Dik Seyyed in Gorgan Plain. All sherds of context 9 in Tr4, dated to 6600 BC, are CNS type.

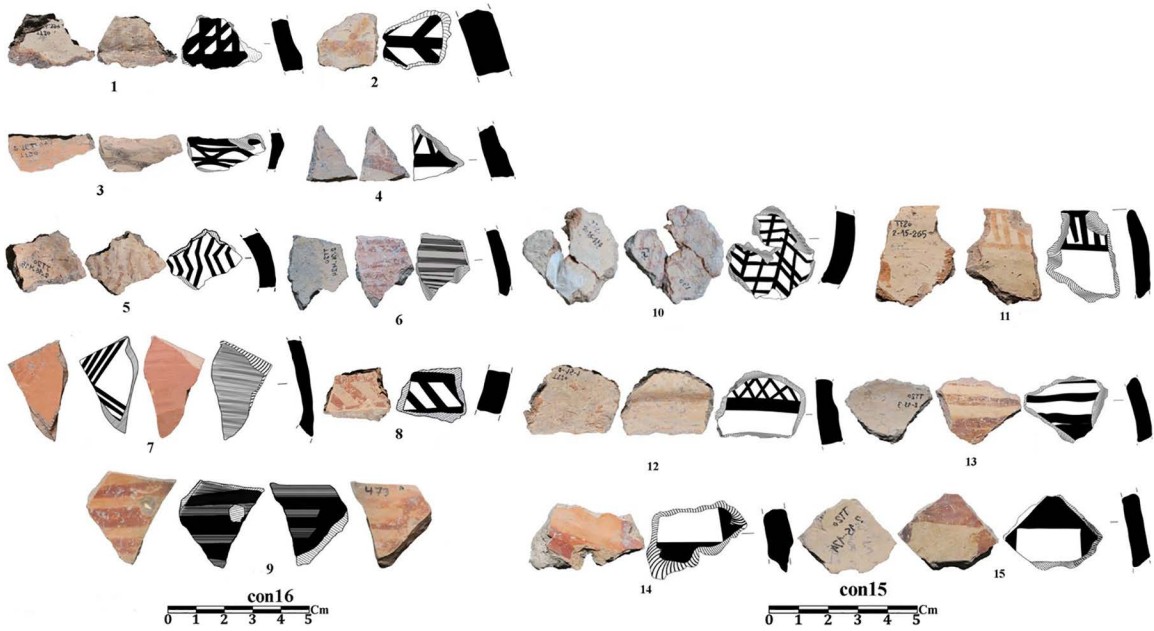

**Fig 7. Neolithic sherds of Touq Tappeh, trench TT2, Con 15 and 16.**

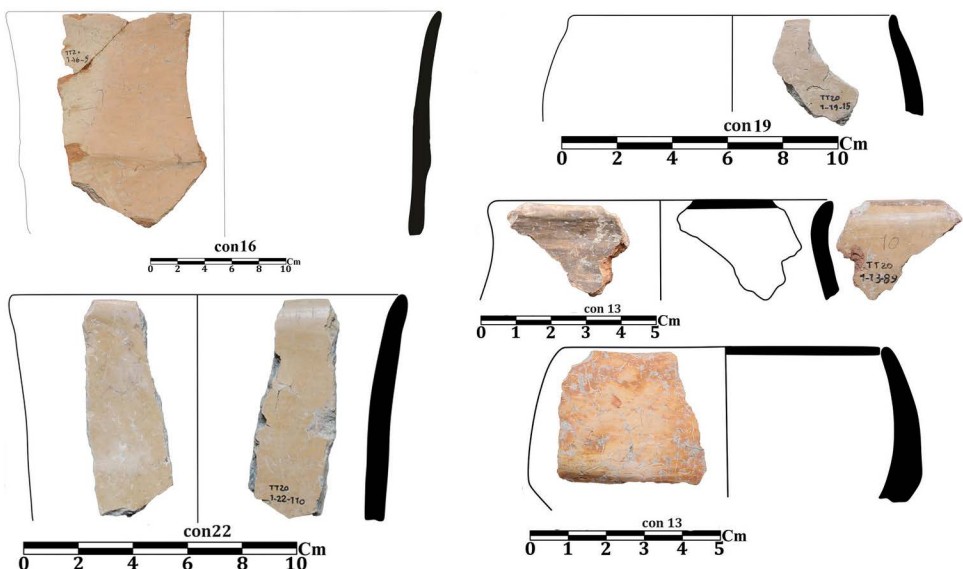

**Fig 8. Neolithic sherds of Touq Tappeh, trench TT1.**

The recently surveyed sites, Tappeh Fakhi and Muzaffar Tappeh in eastern Mazandaran Plains have a few sherds with Djeitun culture features. In more than 40 surveyed and excavated sites on the Gorgan Plain, the Djeitun type has been reported, which Tappeh Bandar Gaz is the nearest one to Tappeh Fakhi, 30 km away; therefore, it is very likely that the Djeitun pottery type entered the eastern Mazandaran through the Gorgan plain. In eastern Mazandaran highlands in which there are no signs of the Djeitun culture, the only typical Neolithic sherds are the CNS type [3]. Therefore, regional

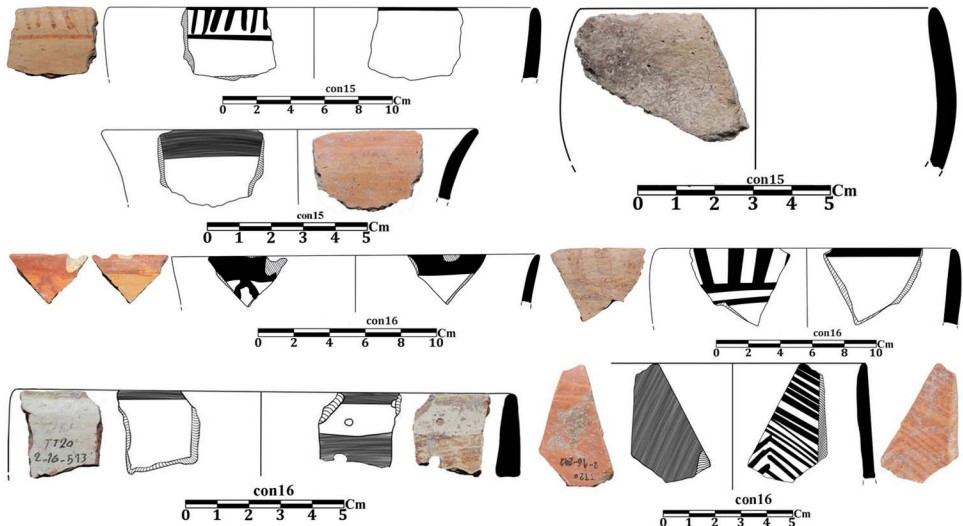

**Fig 9. Neolithic sherds of Touq Tappeh, trench TT2.**

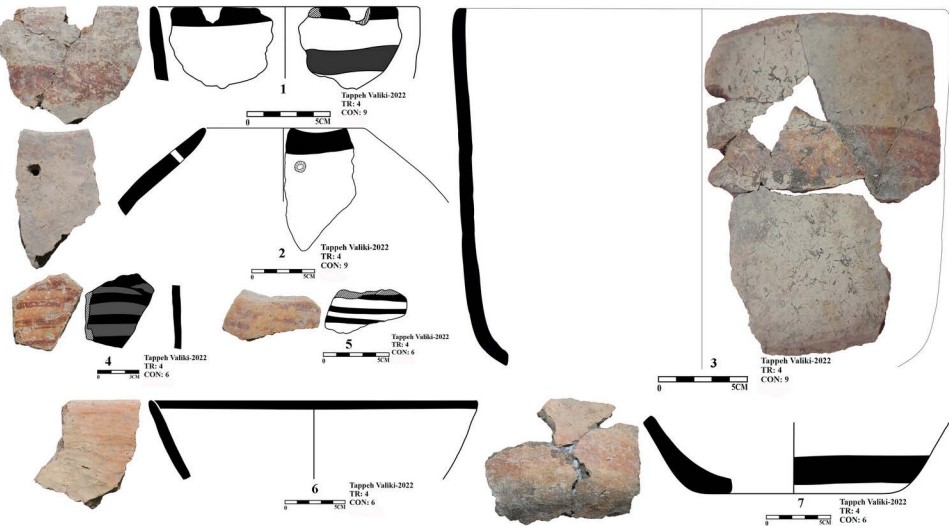

**Fig 10. Neolithic sherds of Tappeh Valiki, trench Tr4; the CNS type (1-3 and 6-7); Djeitun type (5-4).**

connections of eastern Mazandaran (between plains and highlands) are relatively strong due to finding CNS type sherds. Meanwhile, a limited inter-regional connection between the plains and the Djeitun culture through the Gorgan Plain can be observed. Therefore, further excavations and field surveys, especially in the Mazandaran highlands, will be promising to confirm this hypothesis.

Neolithic painted sherds from Touq Tappeh and Tappeh Valiki can be divided into two mains groups. The first group is the CNS type, which is known as *regional pottery*. The majority of the sherds are simple and unpainted, however, most motifs are colored bands on the rim and body, as well as a limited number of complex geometric motifs. The "*Decorative Outer Slip*" (the DOS) is also used to decorate the sherds [3]. The age of 6400–6300 BC is suggested for this group based

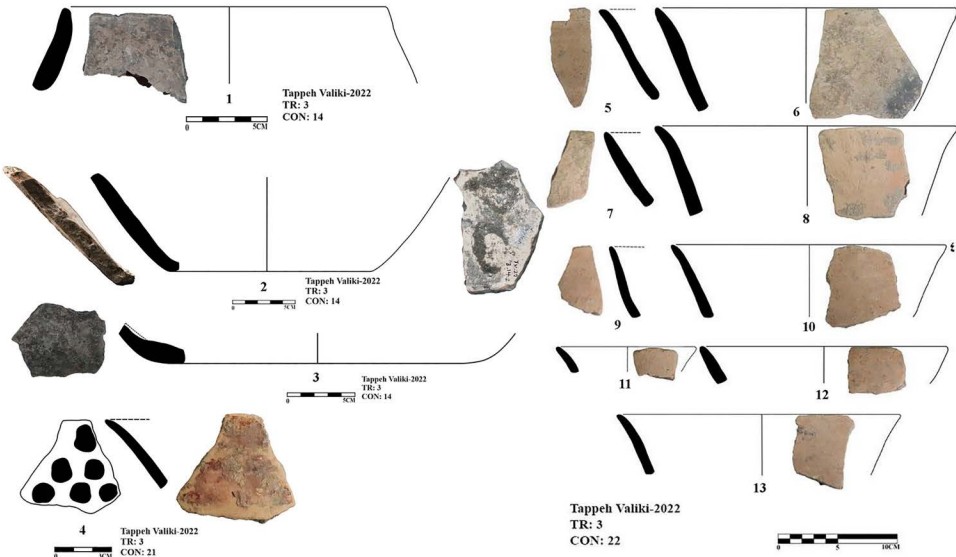

**Fig 11. Neolithic sherds of Tappeh Valiki, trench Tr3; the CNS type (1-13); Djeitun type (4).**

on the recent dating from Hotu Cave; however, the dating of the Tappeh Valiki has taken the CNS culture back almost 200 years, to 6600–6450 BC. Another motif, which previously thought to be found only in plains and known as *local pottery*, is characterized by *horizontal ladder* on the rim. This type of pottery has been observed in contexts 15 and 16 of TT2 at Touq Tappeh with great diversity (Fig 7, sherds 8, 11; Fig 9, sherds 1, 6). Also, it has been recovered from contexts 3 and 6 of Tr4 in Tappeh Valiki. This group is dated 6000–5800 BC. However, we found a similar sherd on Hotu`s pottery assemblage from Coon`s excavations by searching in the photos provided by Christopher P. Thornton.

As mentioned above the CNS is currently considered the oldest pottery in the eastern Mazandaran; it seems that according to the thickness of PN layers at Touq Tappeh and Tappeh Valiki, which indicates a long and continuous occupation, the origin of PN in the region should be sought in the eastern Mazandaran plains. Before the excavations of these sites, Hotu and Kamarband were known as the oldest PN sites; now, based on the absolute dating from Touq Tappeh and Tappeh Valiki, it can be said that pottery production started at least 150 years earlier and with more variety in the plain. This date can probably be older because, in the context 23 in Tappeh Valiki Tr3, we had to stop the excavation due to the rise of underground water. Therefore, it can be assumed that the transition phase from the PPN to the PN can be found in the eastern Mazandaran plains.

The second group of sherds are *inter-regional pottery*. These sherds are very limited and have only been found in the excavations of sites located in the plain. The *shady motif*s sherds that were found in Touq Tappeh and Tappeh Valiki, which is one of this inter-regional type and is dated to 6000–5800 BC, have been found at Pookerdvall, Yarim Tappeh, and Aq Tappeh in Gorgan Plain, as well as Deh Kheir and WMSeC in the Shahrud Plain. The date of these sherds is the same as the Early Djeitun (phase I) culture, which belongs to the late 7th and the beginning of the 6th millennium BC. A sherd with *polka motifs* was found in context 21 at Tappeh Valiki; Coolidge in her thesis [1] used to mention this design as *dot motifs*, however the motifs are bigger to be a dot; therefore, we would like to use *polka motifs* which we believe are more proper to mention this design. Similar sherds have been reported in adjacent regions such as the Central Plateau, Gorgan Plain, Bastam Plain, and Djeitun sites in Southern Turkmenistan (Fig 11, sherd 4). In the Djeitun culture classification, this motif is placed in phase 2 and belongs to the mid-6th millennium BC, the same as the dating of Sialk I. We suggest the *polka motifs* in Tappeh Valiki based on the dating of Context 22, which is concurrent to Context 21, belongs to 6450–6300 BC (Fig 12).

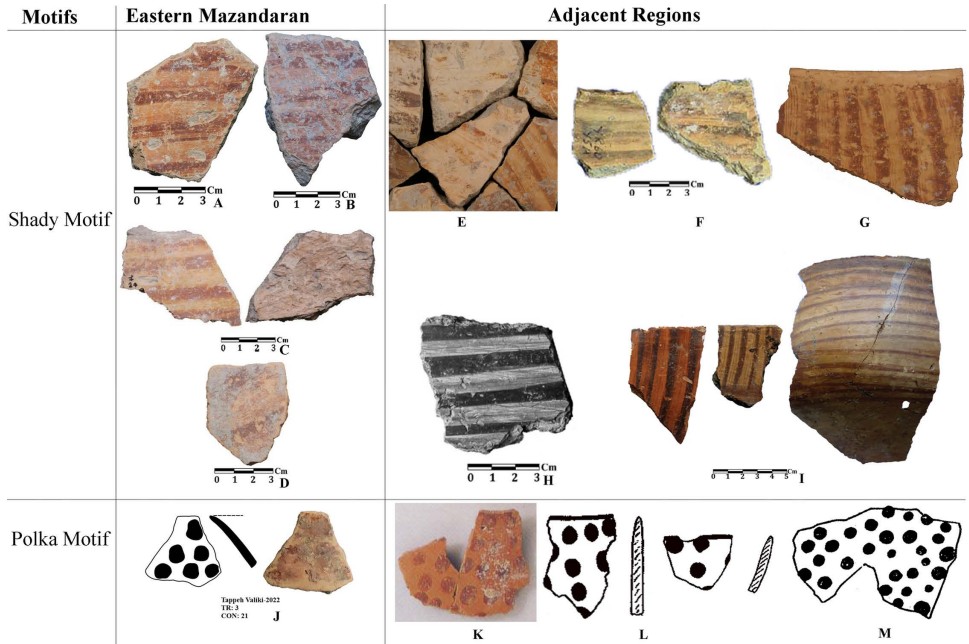

**Fig 12. Inter-regional motives comparing:** A, J) Tappeh Valiki [28]; B) Touq Tappeh [27]; C) Fakhi [3]; D) Muzzafar [3]; E) Pookerdvall [48]; F) Yarim Tappeh [51]; G) Aq Tappeh [49]; H) Deh Kheir [46]; I) the EMSeC [41]; K) the EMSeC [40]; L) Sialk [52]; M) Togolok [1].

In terms of potteries and their motifs, there is not much material evidence regarding the cultural connections between the Neolithic sites in eastern Mazandaran, northeastern Iran, and southern Turkmenistan. However, in the following, we will consider this issue from other perspectives. Another promising point in this regard is the recovery of a sherd from the context 9 Tr4 in Tappeh Valiki, which has a hole near the rim (Fig 10, sherd 2), dated to 6600 BC; this technique has been observed in Touq Tappeh, context 16 of TT2 dated to 6000 BC (Fig 9, sherd 4). Sherds with the same feature, which may be related to hanging, as well as, strapping and repairing vessels, have been reported from Djeitun, Chakmakli, Djebel, and DamDam Cheshmeh in Turkmenistan; Dik Seyyed, Pookredvall, and Aq Tappeh in Gorgan Plain; Sang-e Chakhmaq in Bastam Plain, and Pahlavan in North Khorasan. All these sites belong to the late 7th and early 6th millennium BC, but the mentioned feature is based on the dating of the context9, belongs to the mid-7th millennium BC, thus, showing the earliest evidence of this behavior (Fig 13).

## 6.2. Lithics

A total of 321 stone artifacts were discovered at Tappeh Valiki which consist of 59 tools, 182 simple debitage pieces, 13 cores, 61 debris, 1 piece of raw material, and 5 unidentifiable items (Fig 14C). Typologically, the tools include the retouched in abondance, multifunction tools, notches, notch-denticulate, and scrapers. Most of the cores are flake-type. Indirect percussion so often, and direct percussion and pressure techniques are rarely used to remove flakes from cores. The most frequently found debitage types include chip, blade, bladelet, chunk, Clactonian and kina, burin spall, and blade debitages; among them, a few sickle sheenblades and bladelets have also been identified.

A high frequency of notch-denticulate and scrapers has been reported in the Neolithic lithic assemblage from Touq Tappeh, while there are few blades and bladelets (Fig 14A, B). A few sickles sheen blades and bladelets recovered from this site as well. Flakes, blades, bladelet cores, and lithic fragments with cortex have not been found; however, debris are in abundance. The sources of raw materials are located approximately 10 km from the site [54]. Therefore, part of the production process was conducted away from the sites. Probably, the lithics of Touq Tappeh indicate a continuation of the Caspian

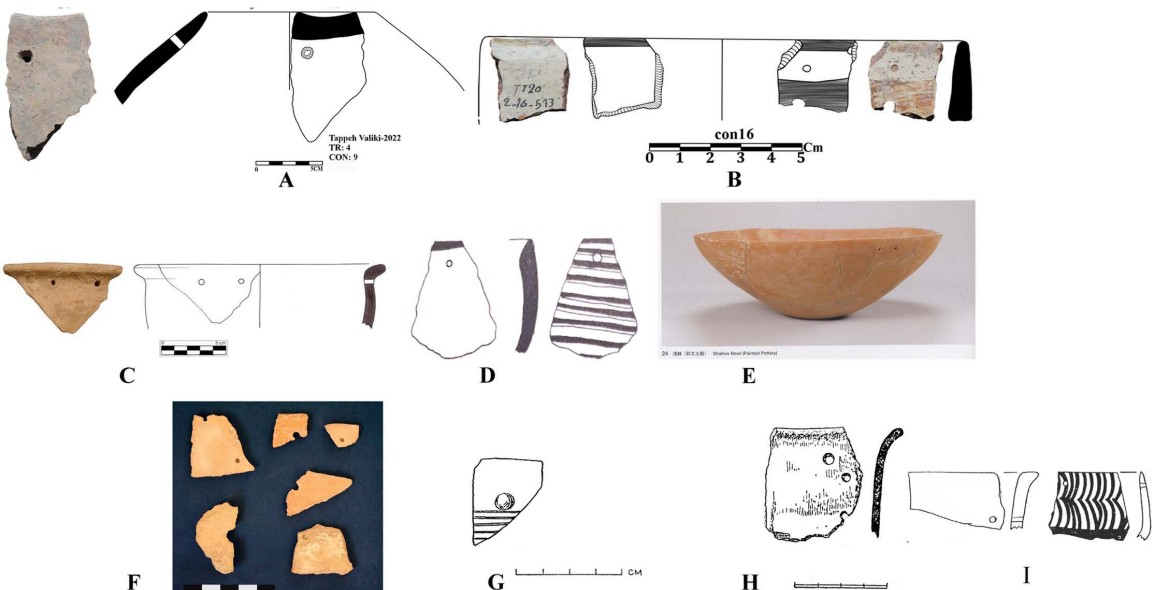

**Fig 13. Comparison of the sherds with holes in the body: A) Tappeh Valiki, 6600 BC [28]; B) Touq Tappeh, 6000 BC [27]; C) Pookerdvall, 6th millennium BC [48]; D) Aq Tappeh, 6th millennium BC [49]; E) the EMSeC, 6000 BC [33]; F) Tappeh Pahlavan, 5300 BC [53]; G) Dam Dam Cheshme, Late Neolithic [1]; H) Djebel, Late Neolithic [1]; I) Djeitun, 6100 BC [1].**

Epipalaeolithic tradition and are comparable to the lithics assemblage from Komishan Cave and are based on flake production [27,55]. However, it seems that blade production started in the late 7th millennium BC in the lithic assemblage of Touq Tappeh, albeit limited. Mozhgan Jayez, by studying the lithic assemblages of Hotu, Kamarband, Kamishan, and Al Tappeh caves, states that since 9000–7000 BC, the pressure technique appeared as a major change in the toolkit of the eastern Mazandaran. This technique was used to make regular and narrow blades and bladelets that were used to harvest crops. She also points out that the rest of the tool types continued without any changes in the following periods. it is still impossible to say whether the pressure technique is autochthonous or imported [56,57] Therefore, the pressure technique, as well as the production of blades and bladelets in the WMSeC, which have been interpreted in relation to the beginning of agriculture [58], had started at least 2000 years earlier in the eastern Mazandaran; on the other hand, mortars and pestles (see in special findings section) were also produced in the Epipalaeolithic of the region much earlier than the WMSeC.

The high abundance of notch-denticulate and scrapers, which were probably used to cut and grind hard objects such as wood and bone, as well as to skin hunted animals, is the main difference between the Touq Tappeh lithic assemblage and other sites in adjacent regions such as Central Zagros, Central Plateau, Shahrud and Bastam plains, and the Djeitun and Kelteminar cultures. The lithic assemblage from Touq Tappeh, dated to the late 7th and early 6th millennium BC, likely indicates a hunter-gatherer community on the eastern Mazandaran plains. The analysis of zooarchaeological data from Hotu has shown that although the management of goat and sheep species was practiced in the PPN and PN, none of them show signs of domestication in terms of genetic morphology [35]. It seems that the eastern Mazandaran communities were still in the transition phase from hunting-gathering to food production in the period from the mid-7th to the beginning of the 6th millennium BC.

## 6.3. Architecture

Due to stratified and vertical excavation, we did not have much evidence of architecture in these two sites (Fig 15); however, in context 22 of Tr3 in Tappeh Valiki, some architectural remains belonging to the PN were unearthed which included circular mudbrick pisé wall extending through the trench's southwestern section. A portion of the trench's wall was

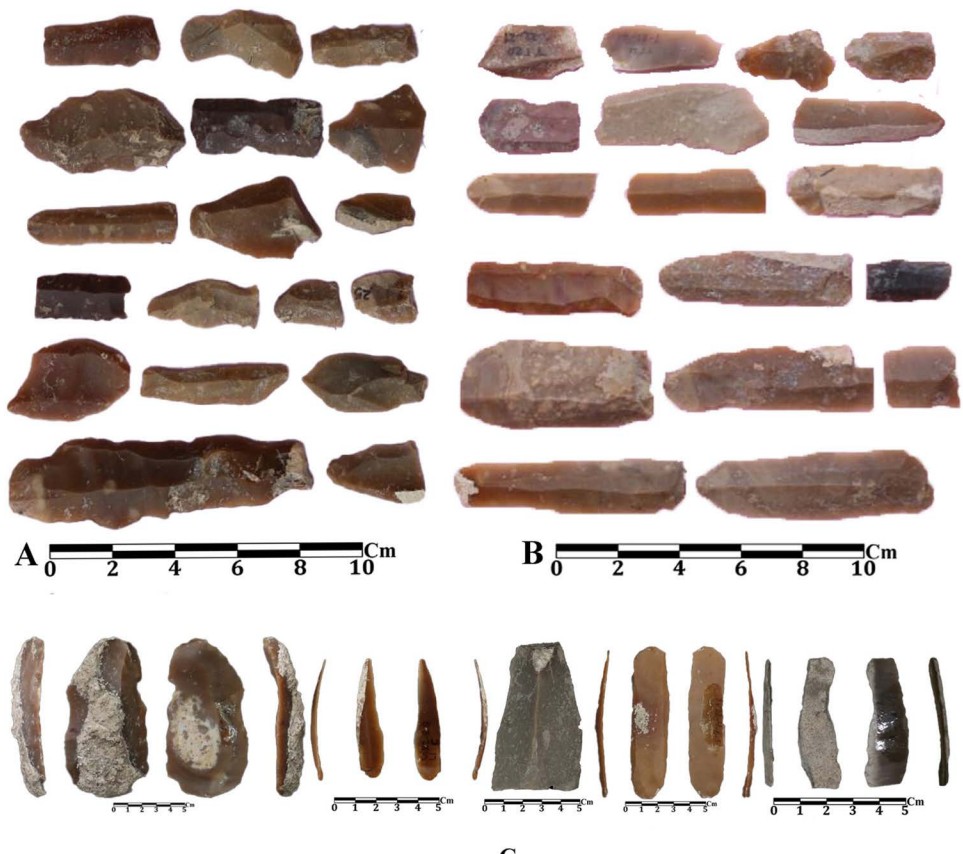

**Fig 14. Lithics from Touq Tappeh (A-B) and Tappeh Valiki (C).**

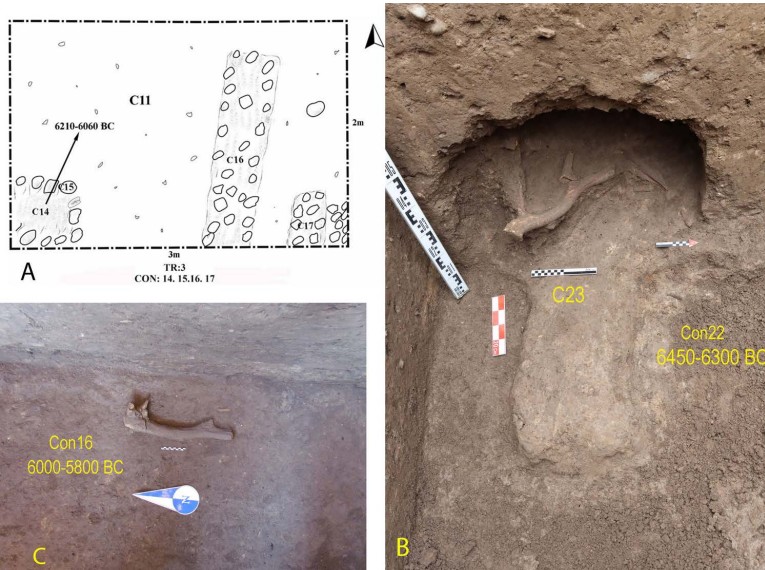

**Fig 15. A and B) architectural remains of Tappeh Valiki, Tr3; C) deer antler in Touq Tappeh, context 16, TT2.**

excavated to reveal these remains better, uncovering animal bones and horns, likely deer antlers, above the architecture remains. Based on absolute dating, the mentioned remains are dated to 6450–6300 BC or even older. Unfortunately, the excavation was unfinished due to the rise of underground water. A deer antler, also, was found in Context 16 of TT2 Touq Tappeh; Although, there was not any architectural space. The positioning of wild animal horns in architectural space at early Neolithic sites (10th to 8th millennium BC) in Central Zagros, such as Sheikhi Abad, Asiab, Ganj Dareh, and Ali Kosh, was analyzed in relation to Neolithization and domestication behaviors [59]. The use of animal horns, especially deer antlers, can be seen in the Neolithic sites in northeastern Iran, such as Baluch, Deh-Khair, and EMSeC, dating to the late 7th and early 6th millennium BC; note that a deer antler was also found from the Epipalaeolithic layers of Hotu which indicate the origins of this behavior in older periods of eastern Mazandaran (Fig 16).

Context 15 of Tr3 in Tappeh Valiki is the remains of a heart; inside this heart is marked as context 14 and is concurrent to contexts 16 and 17 which are stone walls. The use of stone foundations with mud mortar has been observed in Neolithic sites such as Kalateh Khan, dated 5600–5300 BC [47]. The dating sample from inside the oven (context 14) belongs to 6200–6100 BC, which can be considered for contexts 16 and 17 as well. In the adjacent regions, the oldest ones belong to 7200–6800 BC at the WMSeC. However, at the EMSeC, architectural remains dated to 6200–5700 BC [40]. Therefore, at present, Tappeh Valiki probably shows the oldest architectural remains of PN period in the east of Mazandaran, and in an even wider area, the northeastern of Iran.

## 6.4. Special finds

In the excavations of the PN contexts of Tappeh Valiki and Touq Tappeh, a few, but special, objects have been obtained. These finds are made of different materials and have various uses. Although they are few, they probably indicate a cultural connection with regional and inter-regional Epipalaeolithic and Neolithic sites. The investigation of these objects indicates that they mostly show regional connection and cultural continuity from the Epipalaeolithic to PN, and besides that, they also indicate connection with the adjacent regions (Figs 17 –20).

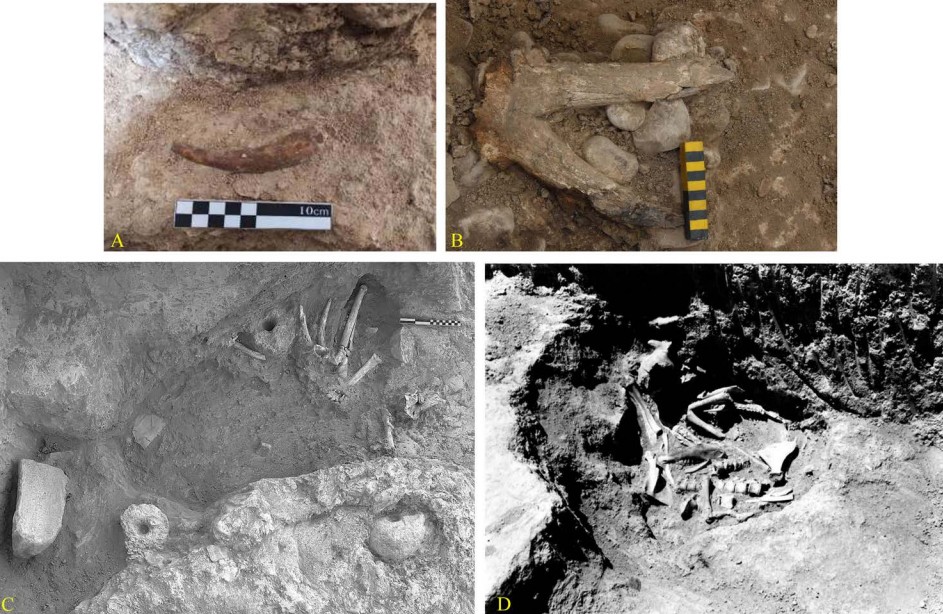

**Fig 16. The use of deer antlers in architectural spaces and sites: A) Hotu Cave (Epi-Paleolithic) B) Baluch (6th millennium BC) C) Deh Kheir (6th millennium BC) D) The EMSeC (end 7th to mid-6th millennium BC).**

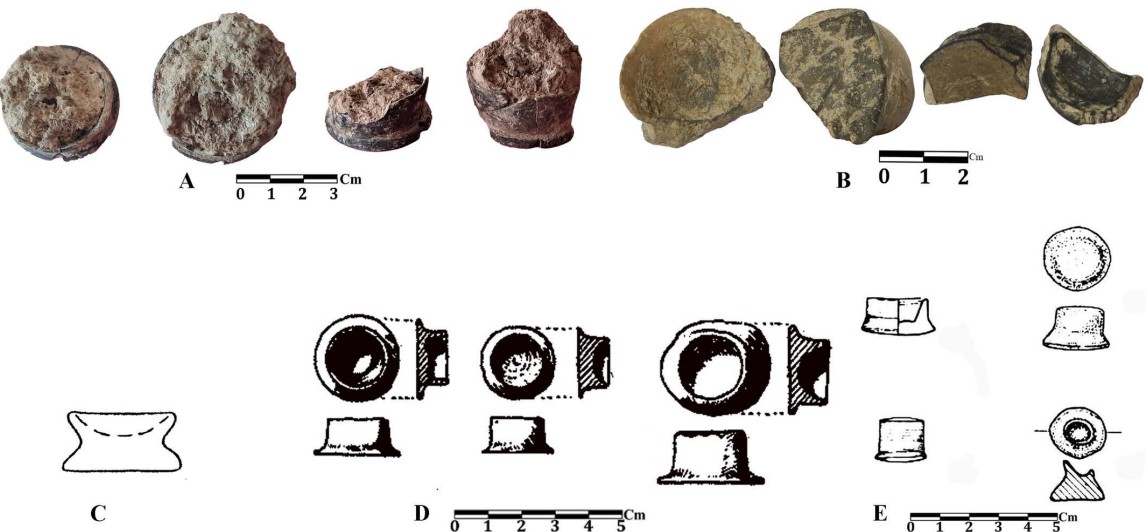

**Fig 17. Comparison of small containers: A) Tappeh Valiki, 6600 BC [28]; B) Touq Tappeh, 6000 BC [27]; C) Djeitun, 6100 BC [1]; D) Sialk I, 5600 BC [52]; E) the EMSeC, 6000 BC [39].**

One of the most interesting categories of objects is small (or miniature) containers (Fig 17). These small containers have an average diameter between 1.5 and 2 cm and a height of 2 cm. Their depth is one centimeter or less. All of the containers have smooth black slip, except for one which has brown-cream slip and mineral temper. Fig 17 summaries similar vessels reported from the Late Neolithic sites of adjacent regions such as Djeitun and Chagily in Turkmenistan, the EMSeC (level 3) in Shahrud Plain, and Sialk I in Kashan. Similar small containers have been found in South Zagros, at sites belonging to the Mushki phase such as Tappeh Rahmat-Abad, Tall-e Mushki, Tol-e Nurabad, and Tol-e Bashi [[60,53,61]53]. The use of these vessels is unclear; however, a few archaeologists have mentioned them as cosmetics vessels [62,52,39]. These objects have not been reported in excavations at the caves of eastern Mazandaran.

A polished deer antler obtained from context 6 of trench Tr4 at Tappeh Valiki (6000–5900 BC) is comparable to similar ones from the old and new excavations of the Kamarband cave from Epipalaeolithic layers [6,30]; Only one bone tool was found from the context of 16, trench TT2 of Touq Tappeh (6000–5800 BC). This bone object is probably a chisel or a needle, and similar finds have been reported from almost all Epipalaeolithic and Neolithic sites in the region and beyond [1,63,64,39,46] (Fig 18).

Three baked clay objects, including a spindle whorl, a disk, and probably a small pestle, were obtained from Touq Tappeh TT1, dated 6250–6000 BC. These clay objects are comparable to those from PN sites in the adjacent regions, while such objects have not been reported in any of the past and recent excavations of caves in the eastern Mazandaran. A brown-colored polished clay spindle whorl has been recovered from Context 17 (late 7th millennium BC); it may indicate using livestock wool and hair since the management of goat and sheep was started in the PPN [35]. Similar objects have been reported at Baloch ([65]), Djeitun [1], and the EMSeC [39]. The clay disc from Context 22 (6250–6100 BC) has a polished surface and a creamy slip similar to the Neolithic sherds in this site; In the EMSeC [39] and Djeitun culture sites, including Djeitun itself [1], similar objects are mentioned. However, due to the low quality of the drawings and the lack of proper description, it is impossible to compare them. The last clay object has a conical shape and thick creamy-yellow slip (similar to the clay disc). The use of this object is unclear, though, according to the recovery of small containers at this site (Table 7) it may be a small pestle. This object was recovered from context 20 (6250–6100 BC). Similar objects have also been found in the sounding program of Tappeh Valiki [28], the EMSeC [41], Djeitun [1], and the field survey of Narges Keti site [31] (Fig 19).

A few stone objects were found, likely related to the processing of plants. The most notable find is a conical-shaped pestle, discovered in context 6, trench Tr4 at Tappeh Valiki (6000–5900 BC), which is similar to the one reported from

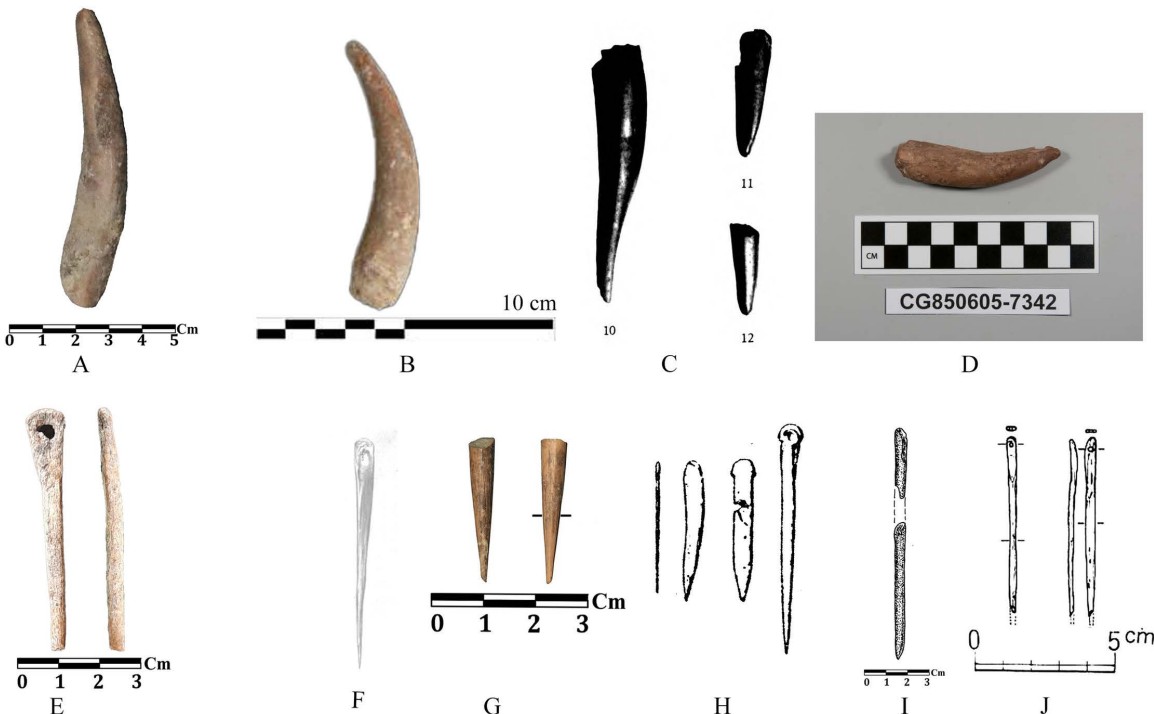

**Fig 18. Antler and bone objects of Touq Tappeh and Tappeh Valiki an comparing with adjacent sites: A) Tappeh Valiki; B) Kamarband (recent excavation); C) Kamarband (old excavation); D) Hotu; E) Tappeh Valiki; F) Hotu (old excavation); G) Al Tappeh; H) Djeitun; I) Deh Khier; J) The EMSeC.**

the Epipalaeolithic layers of Hotu Cave. Both pestles are made of limestone and have the same shape and size. Also, similar pestles have been found at Komishani open site [25] and the Kamarband Cave [6] from the Epipalaeolithic period. Therefore, these pestles can indicate a cultural continuity from the Epipalaeolithic to the PN in the eastern Mazandaran region. There are pestles with similar shapes at the WMSeC [39] and Djeitun [1], although, the pestle at the WMSeC was made of sandstone and the one at the Djeitun lacks photos or descriptions. Stone objects, probably small mortars, have been recovered from context 18, trench TT1 at Touq Tappeh (late 7th millennium BC). In the middle part of one of them, the remains of red pigment can be observed, which may suggest that it has been used to crush and grind mineral or plant materials to obtain red color; The red pigment could be used to decorate products such as pottery or human burials. Although no PN burial has yet been found to show this cultural continuity between Epipalaeolithic and the PN, there is evidence of using red pigment for human burials in Kamarband Cave excavation in the late Epipalaeolithic (levels 19–21) [6]; in the WMSeC, a mortar with traces of red material was reported [39]. The last stone object is a broken circular disc with a hole in the middle; This object was found in context 9, trench Tr4 of Tappeh Valiki (6600–6400 BC). A specific use cannot be assigned to this stone object; however, for a similar object in the WMSeC, a stone spindle whole is considered [41]. In his report, Masuda also mentioned a similar object on the WMSeC and suggested that this stone object is a type of grinding stone [39]. This object was also reported from the PN layers of the Kamarband Cave (level 7 and level B) [1] (Fig 20).

## 7. Conclusion

The dating of Touq Tappeh and Tappeh Valiki, identified as the earliest PN sites on the eastern Mazandaran plains, has significantly contributed to clarifying the PN chronology of the region. Based on current evidence, it can be asserted with greater confidence that the CNS culture in eastern Mazandaran, particularly on the Neka plain, with the oldest radiocarbon

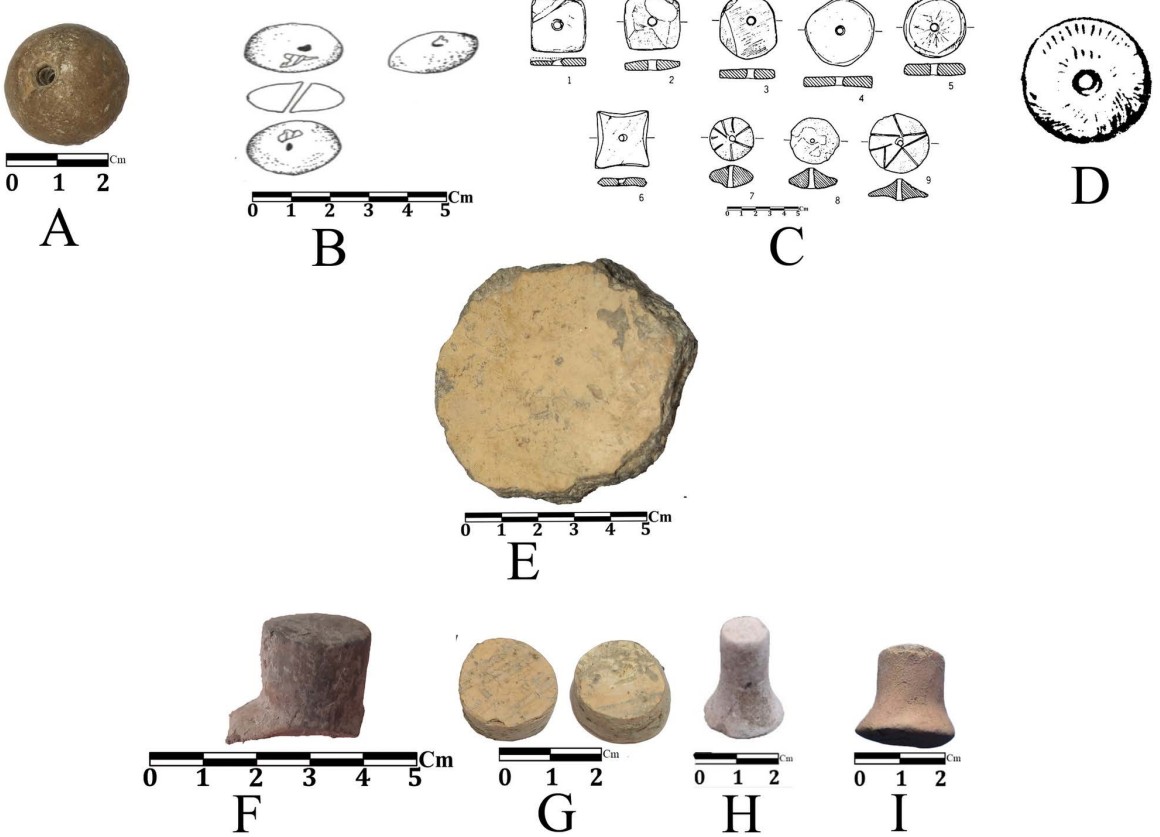

**Fig 19. Clay objects of Touq Tappeh and Tappeh Valiki an comparing with adjacent sites: A, E, G) Touq Tappeh; B) Tappeh Baluch; C, I) the EMSeC; D) Djeitun; F) Tappeh Valiki; H) Narges Keti.**

dating 6600–6400 BC, represents the earliest PN period in northeastern Iran. While the origins of PN societies remain uncertain, the revised chronology has posed a substantial challenge to the Aceramic/Proto-ceramic Chakhmaq hypothesis. This study aims to investigate regional and inter-regional interactions through the analysis of material culture recovered from the excavations at Touq Tappeh and Tappeh Valiki.

The study of more than 2700 Neolithic pottery sherds from the Touq Tappeh and Tappeh Valiki has shown that most of these sherds can be categorized under the CNS culture, which was produced in the region since 6600 BC, and only a small number of sherds from 6000 to 5800 BC layers are of Djeitun/Chakhmaq types. The pressure technique for making blades and bladelets, which was thought to have entered the WMSeC from the central Zagros and then spread throughout northeastern Iran and Central Asia, has now been dated to the 9[th] millennium BC (late Epipaleolithic and during PPN) by this new research, which at least is 1000 years earlier than the WMSeC; in addition, the study of the lithic assemblage from Touq Tappeh indicate the continuation of the Caspian Mesolithic (Epipalaeolithic) tradition and the limited production of blades and bladelets started in the late 7[th] millennium BC. The insignificant remains of architecture from Tappeh Valiki do not help much to understand the connections, although, the tradition of using deer antlers in architecture space from this site represents the oldest evidence of this practice in northeastern Iran, and even before that has also been observed in the Hotu Epipalaeolithic layers. Bone and stone objects, on the one hand, show a strong regional connection, however, on the other hand, clay objects, especially small containers, are comparable to the PN sites on the central plateau, northeastern Iran, and southern Turkmenistan.

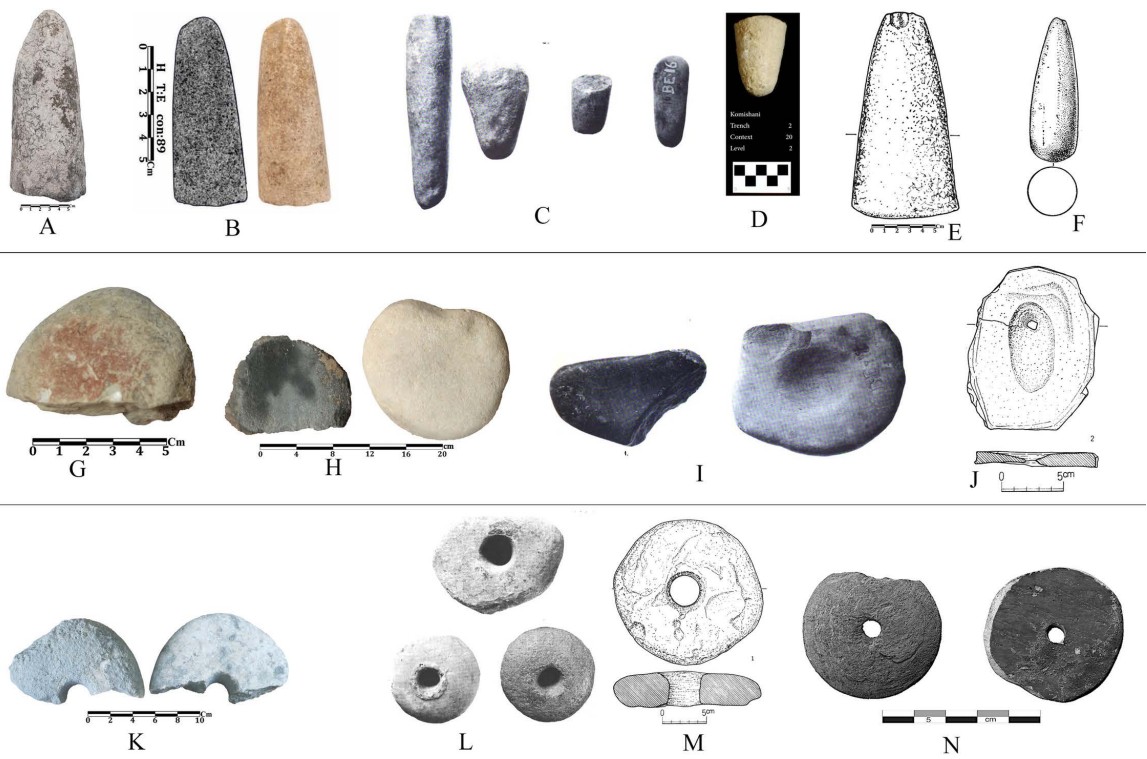

**Fig 20. Grinding stone objects of Touq Tappeh and Tappeh Valiki, and comparing with adjacent sites: A, H, K) Tappeh Valiki; B) Hotu; C, I, L) Kamarband (old excavation); the EMSeC; D); Komishani open site; E, J, M) the WMSeC; F) Djeitun; G) Touq Tappeh; N) Kalateh Khan.**

According to the available evidence, it seems that the PN communities of eastern Mazandaran during this period, which so far is dated to the first half of the 7th millennium BC, appear to have maintained cultural continuity with earlier Epipalaeolithic and PPN traditions; By the end of the 7th millennium BC, limited evidence of inter-regional interactions with adjacent regions is attested through materials such as small containers and clay objects and through a small number of Djeitun/Chakhmaq sherds. Field surveys and the re-examination of pottery assemblages from previous excavations in the highland regions have yielded no evidence for the presence of Djeitun/Chakhmaq type. Consequently, it is highly probable that these materials were introduced into eastern Mazandaran through the Gorgan Plain, suggesting a potential route of cultural exchange.

## Acknowledgments

We would like to thank the University of Mazandaran for corporation to receiving permission for the archaeological excavations at Touq Tappeh and Tappeh Valiki from following organization. We extend our appreciation to the relevant authorities for their support. We are thankful to the Research Institute of Cultural Heritage and Tourism, as well as the Iranian Center for Archaeological Research, for granting permission for this research endeavor. The dedicated and sincere cooperation of the officials from the Cultural Heritage, Tourism, and Crafts Organization of Mazandaran is gratefully acknowledged. We would like to express our deepest gratitude to Keni Zheng, Department of Archaeology, University of Reading, for proofreading the paper and Soghra Ramzi, Department of Plant Biology, University of Mazandaran, for helping us creating the map.

## Author contributions

**Conceptualization:** Rahmat Abbasnejad Seresti, Seyyed Kamal Asadi Ojaei.

**Formal analysis:** Xinying Zhou.

**Writing – original draft:** Rahmat Abbasnejad Seresti, Seyyed Kamal Asadi Ojaei.

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
