## [Decision Letter · Decision Letter 0]

19 Jan 2025

Dear Dr. Abbasnejad Seresti,

Thank you for submitting your manuscript to PLOS ONE. After careful consideration, we feel that it has merit but does not fully meet PLOS ONE’s publication criteria as it currently stands. Therefore, we invite you to submit a revised version of the manuscript that addresses the points raised during the review process.

We look forward to receiving your revised manuscript.

Kind regards,

Joe Uziel

Academic Editor

PLOS ONE

3. In your manuscript, please provide additional information regarding the specimens used in your study. Ensure that you have reported human remain specimen numbers and complete repository information, including museum name and geographic location.

For more information on PLOS ONE's requirements for paleontology and archeology research, see https://journals.plos.org/plosone/s/submission-guidelines#loc-paleontology-and-archaeology-research.

4. Please include a complete copy of PLOS’ questionnaire on inclusivity in global research in your revised manuscript. Our policy for research in this area aims to improve transparency in the reporting of research performed outside of researchers’ own country or community. The policy applies to researchers who have travelled to a different country to conduct research, research with Indigenous populations or their lands, and research on cultural artefacts. The questionnaire can also be requested at the journal’s discretion for any other submissions, even if these conditions are not met.  Please find more information on the policy and a link to download a blank copy of the questionnaire here: https://journals.plos.org/plosone/s/best-practices-in-research-reporting. Please upload a completed version of your questionnaire as Supporting Information when you resubmit your manuscript.

“Funding for the archaeological excavation at Touq Tappeh was generously provided by the University of Mazandaran and the research grant of one of the authors, R. Abbasnejad Seresti. We extend our appreciation to the relevant authorities for their support. We extend our appreciation to the relevant authorities for their support. We are thankful to the Research Institute of Cultural Heritage and Tourism, as well as the Iranian Center for Archaeological Research, for granting permission for this research endeavor. The dedicated and sincere cooperation of the officials from the Cultural Heritage, Tourism, and Crafts Organization of Mazandaran is gratefully acknowledged. Special thanks to Soghra Ramzi for helping us creating the map.”

7. We note that Figures 3 and 4 in your submission contain [map/satellite] images which may be copyrighted. All PLOS content is published under the Creative Commons Attribution License (CC BY 4.0), which means that the manuscript, images, and Supporting Information files will be freely available online, and any third party is permitted to access, download, copy, distribute, and use these materials in any way, even commercially, with proper attribution. For these reasons, we cannot publish previously copyrighted maps or satellite images created using proprietary data, such as Google software (Google Maps, Street View, and Earth). For more information, see our copyright guidelines: http://journals.plos.org/plosone/s/licenses-and-copyright.

a. You may seek permission from the original copyright holder of Figures 3 and 4 to publish the content specifically under the CC BY 4.0 license. 

Reviewers' comments:

Reviewer's Responses to Questions

**Comments to the Author**

1. Is the manuscript technically sound, and do the data support the conclusions?

Reviewer #1: Yes

Reviewer #2: Yes

2. Has the statistical analysis been performed appropriately and rigorously?

Reviewer #1: Yes

Reviewer #2: Yes

3. Have the authors made all data underlying the findings in their manuscript fully available?

Reviewer #1: Yes

Reviewer #2: Yes

4. Is the manuscript presented in an intelligible fashion and written in standard English?

Reviewer #1: Yes

Reviewer #2: No

Reviewer #1: In Line 205 and 231, abbreviations WMSeC and EMSeC are used. These are not explained.

In Line 319 the author uses the trem "dot motif" to describe examples in Fig 12:4. Dot motif is a specific term for a specific decorative scheme from southwestern Iran. The examples referred to in Fig 12:4 are decorated with large, round blobs that may best be described as "polka dot" motif.

Tell is that Arabic version of Persian Tall, hill. Local version of this in Fars is pronounced Tol. So, the name of the sites in line 424 should be spelled as Tol-e Bashi, not Toli-e Bashi and Tell Mushki should be changed to Tall-e Mushki.

In Fig. 3, Cheshmeh Ali is misspelled Chwshmeh Ali, should be corrected.

Reviewer #2: This is an important piece of research which deserves to be fully published. There are many errors in the written English which need to be corrected before this article is acceptable. The topic is significant and the methods and interpretations are all clear, apart from the language issue.

**Do you want your identity to be public for this peer review?** For information about this choice, including consent withdrawal, please see our Privacy Policy

Reviewer #1: **Yes: ** Abbas Alizadeh

Reviewer #2: No

---

## [Author Response · Author response to Decision Letter 1]

17 Mar 2025

Answers to editors:

We wish to show you our gratefulness for addressing the issues in the paper. We tried our best to fulfil your requests and suggestions.

1. As recommended and after reading the Plos One template pdf, we correct the paper styles to the journal`s requirement.

2. As required by the editor we provide information of the organization who gave permission to us for field works.

3. We did not used any human, plant our animal or genetic samples in our paper, however, any other data that required by editor is available.

4. After carefully read the online page, we believe our paper does not need PLOS’ questionnaire on inclusivity in global research given that our research does not include the condition that required by this section at the journal.

5. As you required, we changed the acknowledgment as the way that the journal required. The statement “The author(s) received no specific funding for this work” is correct and we also will mention the current statement in the cover letter as well. The first author (also the corresponding author) received funds from the University of Mazandaran and his research grant for only the fieldwork; the funding does not cover the publication fee.

6. We acknowledge that we agreed to let access to the data by the journal if required. We would like to state that ‘all relevant data are within the paper’.

7. After checking the permission in Google earth website, we notice that the mentioned figure does not need permission from Google earth.

https://about.google/brand-resource-center/products-and-services/geo-guidelines/#google-earth

Also, the “figure 3” was actually a GIS map, not a satellite image, however, we change the map to another one to avoid any issue.

8. As you required, we did check the reference list again and we can assure you all the references are currently available to use.

Answers to reviewers:

We are very thankful for your time and effort to read our manuscript and trying to help us for to improve it. We also are very grateful for your positive comments and understanding the importance of our paper; therefore, we tried our best to fulfil your requests and suggestions.

Reviewer #1

Thank you so much for your time and effort we

1. Thank you for noticing this issue. We would like to mention that the terms “WMSeC” refers to “Western Mound of Sang-e Chakhmaq” and “EMSeC” refers to “Eastern Mound of Sang-e Chakhmaq”. We used these abbreviations to avoid the long name of the site. Both these abbreviations were mentioned in manuscript at the first appearance.

2. Thank you for carefully observation the “dot motif” in pottery section and you are right; these round bubbles like motif are too large to be considered as dot and we notice it as well. However, because in her thesis, Coolidge, named it dot motif we tried to be loyal to the original name. Although what we will do to satisfy your require is to let readers know that the name Coolidge given to these motives are “dot motif” but we think it is more proper to call it “Polka motif”.

3. Thank you for reminding us about the sites name that we incorrectly put “Tell”. This is very important point and we will change it to the correct version as you recommended.

4. Thank you for mentioning the misspelled. We correct it.

Reviewer #2

Thank you so much for your time and effort. We understand your concern about the language of the paper; therefore, we asked scholars at British universities to revise the language of the paper.

---

## [Editor Report · Decision Letter 1]

23 Mar 2025

The Oldest Pottery Neolithic (PN) Culture of Northeastern Iran: First Absolute Dating from Eastern Mazandaran Plains

PONE-D-24-56302R1

Dear Dr. Abbasnejad Seresti,

We’re pleased to inform you that your manuscript has been judged scientifically suitable for publication and will be formally accepted for publication once it meets all outstanding technical requirements.

Kind regards,

Joe Uziel

Academic Editor

PLOS ONE
---

## [Editor Report · Acceptance letter]

PONE-D-24-56302R1

PLOS ONE

Dear Dr. Abbasnejad Seresti,

I'm pleased to inform you that your manuscript has been deemed suitable for publication in PLOS ONE. Congratulations! Your manuscript is now being handed over to our production team.

Kind regards,

on behalf of

Dr. Joe Uziel

Academic Editor

PLOS ONE